# Double-Edged Sword: Urbanization and Response of Amniote Gut Microbiome in the Anthropocene

**DOI:** 10.3390/microorganisms13081736

**Published:** 2025-07-25

**Authors:** Yi Peng, Mengyuan Huang, Xiaoli Sun, Wenqing Ling, Xiaoye Hao, Guangping Huang, Xiangdong Wu, Zheng Chen, Xiaoli Tang

**Affiliations:** 1College of Animal Science and Technology, Jiangxi Agricultural University, Nanchang 330045, China; 2Institute of Hydrobiology, Chinese Academy of Sciences, Wuhan 430072, China; 3Jiangxi Provincial Key Laboratory of Conservation Biology, College of Forestry, Jiangxi Agricultural University, Nanchang 330045, China

**Keywords:** urbanization, gut microbiota, amniotes, phenotypic plasticity, microbiome-mediated adaptation

## Abstract

Projections indicate that the global urban population is anticipated to reach 67.2% by 2050, accompanied by a threefold increase in urban built-up areas worldwide. Urbanization has profoundly transformed Earth’s natural environment, notably characterized by the drastic reduction and fragmentation of wildlife habitats. These changes contribute to local species extinction, leading to biodiversity loss and profoundly impacting ecological processes and regional sustainable development. However, within urban settings, certain ‘generalist’ species demonstrate survival capabilities contingent upon phenotypic plasticity. The co-evolution of gut microbiota with their hosts emerges as a key driver of this phenotypic plasticity. The presence of diverse gut microbiota constitutes a crucial adaptive mechanism essential for enabling hosts to adjust to rapid environmental shifts. This review comprehensively explores amniote gut microbial changes in the context of urbanization, examining potential drivers of these changes (including diet and environmental pollutants) and their potential consequences for host health (such as physiology, metabolism, immune function, and susceptibility to infectious and non-infectious diseases). Ultimately, the implications of the gut microbiome are highlighted for elucidating key issues in ecology and evolution. This understanding is expected to enhance our comprehension of species adaptation in the Anthropocene.

## 1. Introduction

Human activities are widely recognized as the dominant drivers of global environmental change [1,2]. Among these, urbanization, characterized by an increased share of the urban population and the physical expansion of cities within a country, represents one of the most extensive anthropogenic transformations of land use [1,3]. Research has indicated that only approximately 40% of the current built environment predates 1975, highlighting its unprecedented pace [1,4]. Projections indicate that 68% of the global population will reside in urban areas by 2050 [5]. This rapid expansion profoundly alters ecosystems through the conversion of natural and agricultural lands to industrial, residential, and commercial zones, driving habitat loss, fragmentation, pollution escalation, and microclimatic shifts. These changes collectively fundamentally restructure ecosystem functions and community dynamics [1,6,7].

Urbanization reconfigures landscapes, resulting in notable changes to Earth’s natural environment, such as a drastic reduction and fragmentation of wildlife habitats. This leads to novel selective pressures on local species, reducing biodiversity and profoundly affecting ecological processes and regional sustainable development. As cities encroach upon biodiversity hotspots, they disrupt species assemblages and modify critical ecological interactions—including predator–prey relationship networks [1,8,9]. Consequently, urban expansion leads to the extinction of a large number of wildlife species, significantly contributing to global biodiversity loss [10,11,12]. Notwithstanding these challenges, urbanization creates novel niches that favor a limited cohort of “versatile” species that necessitate rapid behavioral and physiological adaptations [12,13]. While urban environments provide alternative resources and refuge, facilitating coexistence with humans [5,13,14,15], they concurrently increase exposure to anthropogenic stressors—including pollutants, human–wildlife conflict, and emerging pathogens. Species exhibiting ecological and phenotypic plasticity, a generalist diet, and high dispersal capacity demonstrate greater success in urban environments [16,17]. For instance, coyotes (*Canis latrans*) and synanthropic birds (e.g., pigeons, sparrows) thrive through modified foraging strategies, nesting behaviors, and human interactions [13,14,18].

The gut microbiota (GM) is a complex community of microorganisms that inhabit the digestive tract, playing essential roles in nutrient metabolism, immune modulation, and host health [19]. It has recently emerged as a pivotal mediator of animal responses to environmental change [1,3,20,21]. Host-associated microbial communities, particularly within the gut, are integral to digestion, nutrient assimilation, immune regulation, xenobiotic detoxification, and pathogen defense [22,23]. GM composition is shaped by host-intrinsic factors (e.g., genetics, physiology) and extrinsic variables (e.g., diet, pollutant exposure, habitat characteristics), rendering it a critical indicator of urban adaptation [3,21,22,24]. Furthermore, emerging evidence suggests microbiota plasticity facilitates rapid phenotypic adjustments, indicating GM may be a vital adaptation mechanism critical for promoting host adaptation to rapid environmental changes [25,26,27], which potentially buffers hosts against urban stressors [3,21,28,29].

However, current research predominantly documents microbiota changes in individual wildlife species under urbanization, while systematic cross-taxonomic analyses of response variation and mechanistic drivers remain limited—particularly for non-model vertebrates. The co-evolution of the GM with the host under urbanization is a crucial factor driving phenotypic plasticity [30,31], representing an emergent frontier with profound conservation and public health implications. The relationship between urbanization and GM diversity is discussed in the following sections (Figure 1): (1) the characterization of gut microbiome changes in five amniote groups (reptiles, birds, carnivores, non-human primates, and humans) under urbanization; (2) a synthesis of evidence on how urbanization, diet transitions, and environmental contamination collectively reshape vertebrate wildlife GM; (3) an evaluation how gut microbial alterations in urbanization-adapted amniotes remodel host physiology, behavior, and immune responses; and (4) an analysis of urbanization-mediated impacts on zoonotic pathogen prevalence, antibiotic resistance, and non-communicable diseases (NCDs) via microbiota modulation. By elucidating these mechanisms, we identify knowledge gaps and advocate for integrating microbiome perspectives into urban wildlife management, pathogen surveillance, and biodiversity conservation in an increasingly urbanized world.

## 2. Methods

This systematic review, conducted in June 2025, aimed to assess the response of the amniote gut microbiome to urbanization, diet, and pollution, focusing on implications for zoonotic pathogens, antibiotic resistance, and non-infectious diseases. We performed a systematic literature search across Google Scholar, Web of Science, and PubMed databases without date restrictions, adhering to PRISMA-P (Preferred Reporting Items for Systematic review and Meta-Analysis Protocols) guidelines for study identification, data extraction, and synthesis. The search strategy employed the key terms “gut microbiome” AND (“urbanization” OR “city”) AND (“reptile” OR “bird” OR “carnivore” OR “non-human primate” OR “human”). Identified records were screened against predefined inclusion criteria: (1) subjects must be free-ranging, non-domesticated, terrestrial wild amniotes (excluding captive animals); (2) studies must investigate the gut microbiome in relation to urbanization using 16S rRNA gene sequencing of fecal samples; and (3) studies must directly compare gut bacterial communities between populations inhabiting urban and natural habitats. A single reviewer screened all titles and abstracts, excluding studies failing to meet the criteria. The initial search yielded 870 records, with 35 studies ultimately meeting all inclusion criteria for synthesis (Figure 2).

## 3. Impact of Urbanization on Amniote Gut Microbiota

Host-associated microbiota is integral to host ecology and facilitates wildlife adaptation to rapid environmental changes. Urbanization represents a globally replicated environmental perturbation, providing a powerful framework for investigating wildlife microbiome dynamics. Although microbiome research increasingly highlights the microbiota’s importance in animal hosts facing environmental stressors, studies on urbanization’s impact often focus on single species [1,32,33,34,35,36], limiting generalizable conclusions. Due to intrinsic physiological differences across taxa, findings from one species cannot be directly generalized to others. Here, we synthesized comparative data on GM from urban versus natural habitats across five wild amniotes (reptiles, birds, carnivores, non-human primates, and humans), revealing urbanization-induced alterations in gut microbiomes (Tables 1 and 2). This cross-taxa assessment elucidates the impacts of urban environments and inherent microbial plasticity.

### 3.1. Humans

Urbanization consistently reduces human gut microbial diversity and alters compositional profiles globally, as evidenced by comparative studies between rural non-Western and industrialized [37,38,39,40,41] Western populations from more than 20 regions and 15 ethnic groups between 2013 and 2023 (Table 1). Regarding GM α-diversity, the majority of studies (18/21) report decreased levels in urban-dwelling populations. In contrast, one study demonstrates an inverse trend, attributing this to enhanced food accessibility in urban environments compared to rural settings [42]; whereas two additional studies show no significant differences between rural and urban cohorts, attributable to comparable Western-style dietary patterns in rural populations [43] and specific infant demographics [44]. Key bacteria changes in GM include significant reductions in beneficial fiber-degrading taxa (e.g., *Prevotella* spp., *Treponema* spp., *Xylanibacter* spp.) characteristic of hunter–gatherer microbiomes [45,46] and agricultural communities [39,46,47]. Conversely, urbanization elevates the *Firmicutes/Bacteroidetes* ratio [22,48], increases bile-tolerant *Bacteroidaceae* spp. and *Rikenellaceae* spp. (linked to animal product consumption), and enriches pro-inflammatory genera (*Escherichia* spp. and *Shigella* spp.) along with potentially pathogenic genes [46]. Furthermore, reductions in *Actinobacteria/Firmicutes* [48] and increases in *Bacteroides* spp., *Blautia* spp., and *Coprococcus* spp. (responsive to processed substrates) [22] have been observed from Africa to Asia to the Americas [1,29,49,50,51,52,53,54,55,56,57]. This demonstrates that urbanization can homogenize gut microbiome composition globally, even across genetically distinct populations [33,58].

### 3.2. Non-Human Primates

Non-human primates and their habitats face escalating constraints due to anthropogenic pressures. Research on the impacts of urbanization on the primate GM, however, remains notably scarce. To date, only a single study has explicitly compared gut microbial communities between urban and natural habitats, focusing on Madagascar’s lemurs (*Lemur catta*) [36]. This study documented that heightened disturbance levels restructured lemurs’ gut microbiomes, manifested by diminished microbial alpha-diversity and altered community composition. Critically, these shifts involved a reduction in presumptively beneficial bacteria (e.g., *Bacteroidaceae* spp.) alongside a marginal increase in bacteria associated with dysbiosis (e.g., *Veillonellaceae* spp. representatives) (Table 2).

### 3.3. Carnivores

Similar to non-human primates, only one article compared gut microbial communities between urban and natural habitats [35]. Key findings include the following: (1) urban coyotes consume more anthropogenic food, correlating with increased microbiome diversity; (2) urban coyotes exhibit higher abundances of *Streptococcus* spp. and *Enterococcus* spp., coupled with poorer average body condition. Conversely, rural counterparts harbor microbiomes rich in *Fusobacteria* spp., *Sutterella* spp., and *Anaerobiospirillum* spp., which are linked to protein-rich diets and improved body condition; and (3) diets high in anthropogenic food correlate with increased abundances of *Erysipelotrichiaceae* spp., *Lachnospiraceae* spp., and *Coriobacteriaceae* spp., which associate with larger spleens in urban coyotes (Table 2).

### 3.4. Birds

Urbanization exerts complex, species-specific effects on avian gut microbiomes (Table 2). Most reports (6/8) indicate reduced GM α-diversity in urban-dwelling birds, including species from *Anseriformes* (*Anser cygnoides*), *Passeriformes* (*Zonotrichia leucophrys*, *Geospiza fuliginosa*, *Parus major*, *Passer domesticus*), and *Pelecaniformes* (*Eudocimus albus*). In contrast, attributed to dietary heterogeneity from anthropogenic resources, two species, *Passer montanus* (*Passeriformes*) [59] and *Leptoptilos crumenifer* (*Pelecaniformes*) [60], exhibit increased α-diversity (Table 2). β-diversity analyses consistently distinguish urban and rural communities, with urbanization level strongly predicting compositional shifts [59]. Moreover, consistent alterations in community structure and function are evident. Recurrent taxonomic patterns include decreased *Staphylococcaceae* spp. and increased *Lactobacillaceae* spp. and Proteobacteria in urban birds [29], though phylum-level dominance varies across species. Functional convergence emerges through enriched xenobiotic degradation pathways in urban microbiomes [29], reflecting adaptive responses to anthropogenic contaminants. Species-specific shifts reflect diet/environment/human drivers on avian gut microbiomes. Notably, green spaces within urban matrices mitigate diversity loss, sustaining microbiome richness comparable to rural populations [1].

### 3.5. Reptiles

Research characterizing the intricate and unique symbiotic relationships between reptiles and their gut microbial communities is rapidly expanding. Current studies on urbanization effects are predominantly focused on Squamata, specifically lizards (*suborder Lacertilia*) and iguanas (*suborder Iguania*), identifying ecological factors as key determinants shaping gut microbial diversity across reptilian species.

Research reveals that the gut microbiome diversity of reptiles in urban environments, including lizards such as *Podarcis siculus*, *P. virescens*, *Teira dugesii* [61], *Intellagama lesueurii* [21], and *Bradypodion melanocephalum* [10], exhibits higher diversity compared to conspecifics in natural or semi-natural habitats (Table 2).

### 3.6. Summary

Current evidence demonstrates significant urbanization effects on gut microbiomes across diverse amniotes. However, the impacts vary substantially among taxa, manifesting as increased, decreased, or unchanged α-diversity depending on host species and ecological context (Table 1). Key drivers of these inconsistencies include the following: (1) host-specific living environment; and (2) species-specific physiological (e.g., diet shifts) responses for adaptations. Notably, taxonomic homogenization is frequently observed at multiple levels, including the convergence of GM among humans [43,44] and wildlife [44] across different cities and the convergence of GM between urban-dwelling animals [28], birds [28], carnivores [3], and non-human primates [62] and humans. Critically, expanding human–animal interfaces amplify microbial exchange—historically emphasized for pathogens [37,63,64,65] but increasingly recognized in non-pathogenic transfers that shape microbiome evolution [37,66,67,68]. Collectively, urbanization acts as a multidirectional selective force on the gut microbiome in different amniotes interacting with host ecology to reshape microbial communities in taxon-specific manners [1,28,34,69,70,71,72,73,74,75].

## 4. Key Factors Affecting Amniote Gut Microbiota in Urbanization

Urbanization reshapes animal GM through multiple environmental disturbances, including seasonal variations, regional heterogeneity, climate change, and antibiotic use, with dietary shifts and anthropogenic pollution serving as the primary selective pressures. The transition toward processed foods in urban diets reduces high-fiber food intake, diminishes microbial diversity, and reshapes GM composition and function. Concurrently, pollution-mediated stressors—including water/soil/air pollutants, noise pollution, artificial light, and chemical toxins—induce chronic physiological stress and disrupt microbial communities [76,77,78]. These factors interact with habitat fragmentation, reduced green space accessibility, and host physiological adaptations [78,79]. Collectively, nutritional alterations and anthropogenic contaminants emerge as core drivers of taxon-specific microbial dysbiosis in urban fauna.

### 4.1. Impact of Diet and Nutrition on Amniote Gut Microbiota

Over the past decade, the gut microbiome–diet nexus has garnered increasing attention. Dietary nutrition crucially influences human and animal health through GM mediation. Concurrently, urbanization fundamentally reshapes predator–prey dynamics and nutritional ecology, with gut microbiome plasticity facilitating host adaptation. This process drives distinct dietary shifts in humans and wildlife. As the primary driver of microbiome configuration, the “urban diet” for humans is characterized by increased consumption of processed foods, refined bread, sugars, fats, and meats, alongside a reduced intake of fresh produce and fiber-rich whole grains; conversely, for wildlife, it means increasingly anthropogenic food sources consumption and dietary changes within urban landscapes, with reduced intake of natural prey (Tables 1 and 2). Subsequently, we detail dietary transformations across five amniote taxa and their gut microbial consequences.

#### 4.1.1. Humans

Urbanization drives a global dietary transition characterized by increased consumption of processed foods, red meat, saturated fats, and refined sugars alongside reduced fiber intake from fruits, vegetables, and whole grains—collectively termed the “Western diet” [80,81]. This nutritional shift fundamentally restructures the human gut microbiomes, consistently reducing α-diversity while promoting dysbiosis [80]. The absence of diminished α-diversity in urban populations observed across two studies fundamentally resulted from comparable Western-style dietary patterns in rural and urban cohorts, underscoring diet as a key determinant of gut microbial diversity in urbanization contexts [42,43].

Key microbial alterations include: depletion of fiber-degrading taxa (*Prevotella* spp., *Roseburia* spp., *Ruminococcus bromii*) essential for microbiota-accessible carbohydrate (MAC) metabolism [22,38,39,82]; increased bile-tolerant *Bacteroides* spp., *Oscillospira* spp., and *Alistipes* spp. linked to high-fat/protein intake [1,44,82,83]; and expansion of mucin-degraders (*Akkermansia muciniphila*) during fiber deficiency [49,84]. *Oscillospira* spp. is exclusively detected in the GM of urban populations, likely due to animal-derived diets [17]. This genus thrives under high-bile conditions [85] and relies on microbial fermentation products or host mucus glycans instead of primary fiber degradation [86].

Urban diets also reshape gut fungal communities. The abundance of *Dipodascus geotrichum* spp. and *Hypopichia burtonii* spp. is the highest among the rural population [51]. These environmental fungi are commonly isolated from cereals (corn, wheat, rice). Conversely, urban residents exhibit higher abundances of *Dekkera bruxellensis* and *Hannaella* spp., which are typically associated with fermented products (wine, beer, goat cheese, sourdough). Critically, vertical transmission of GM from mothers to infants via childbirth and breastfeeding perpetuates urbanization-driven microbial diversity loss [87].

#### 4.1.2. Non-Human Primates

Anthropogenic food availability negatively correlates with the *Firmicutes*/*Bacteroidetes* ratio in Japanese macaques (*Macaca fuscata*) [88]. Wild macaques enrich *Firmicutes* (*Lachnospiraceae* spp., *Ruminococcaceae* spp., *Peptococcaceae* spp.)—taxa encoding carbohydrate-active enzymes and metabolic pathways for plant degradation [89]. These microorganisms act as plant degraders with identified key carbohydrate-active enzymes, sugar transport mechanisms, and metabolic pathways. Macaques on anthropogenic diets show lower levels of chloroplasts, reflecting their reduced fiber intake, such as leaves [29]. However, the diversity of GM in macaques is not significantly disturbed by anthropogenic food, possibly owing to omnivorous flexibility and low dependence on high-fiber vegetation [90].

#### 4.1.3. Carnivores

Rural coyotes (*Canis latrans*) harbor higher abundances of *Fusobacteria* spp., *Sutterella* spp., and *Anaerobiospirillum* spp., bacterial taxa involved in amino acid metabolism and typical of carnivores [35,91]. Conversely, urban coyotes show increased *Streptococcus* spp. and *Enterococcus* spp., potentially linked to high carbohydrate intake. As obligate carnivores, coyotes lack efficient carbohydrate digestion, possibly impairing nutrient absorption from anthropogenic food and compromising health [35]. Elevated δ^13^C (determining primary sources of dietary carbon [92]) and reduced δ^15^N (estimating trophic position and protein consumption) in urban coyotes suggest detrimental microbiome alterations despite diversity increases, potentially threatening species survival. In contrast, one study showed that urbanization is a nutritional advantage for endangered San Joaquin kit foxes (*Vulpes macrotis*) and helps their conservation and recovery [93]. Moreover, red foxes (*Vulpes vulpes*), omnivorous species that often feed on anthropogenic food in cities, have a higher survival rate, indicating their adaptation to urban life [94].

#### 4.1.4. Birds

The GM of common birds exhibited rapid adaptation to short-term dietary changes, indicating its plasticity to dietary changes. Urban sparrows, which initially exhibited lower gut microbial diversity, achieved increased diversity after 6 weeks on a rural diet, whereas the opposite effect occurred with urban diets. Generally, low-MAC urban diets (low-fiber, high-fat, processed foods) reduce GM diversity. After 6 weeks of diet intervention, specific microorganisms (*Enterococcaceae*, *Staphylococcaceae*) displayed strong responsiveness to altered diets in house sparrows (*Passer domesticus*). Notably, rural sparrows fed an urban diet showed more pronounced shifts in the composition of these microbial taxa compared to urban sparrows fed a rural diet [29]. This suggests greater adaptability in the rural cohort’s microbiota compared to urban counterparts. Thus, long-term urban diets diminish GM diversity and plasticity, impairing urban hosts’ capacity to benefit from novel (potentially advantageous) diets.

Urbanization alters animal foraging behavior. Herring gulls (*Larus argentatus*) from highly urbanized colonies exhibit heightened site fidelity, shorter foraging trips, and primary reliance on urban food sources. Their GM showed elevated abundances of *Fusobacteria* spp. (linked to carnivory) and *Amaricoccus* spp. (associated with human waste, prevalent in landfills), implying that gull consumption of anthropogenic waste may be more frequent than previously recognized. Conversely, herring gulls breeding in less urbanized areas undertake longer foraging trips. Their dependence on dispersed, stable food sources results in broader foraging ranges—likely explaining the highest GM diversity in minimally urbanized populations [72].

Among scavenger birds, a study comparing the gut microbiota (GM) of marabou storks (*Leptoptilos crumenifer*) feeding on urban slaughterhouse waste and landfill garbage revealed that storks foraging on landfill garbage exhibited significantly higher microbial diversity [60], potentially due to broader organic matter sources. Isolation of *Vagococcus* spp. from the slaughterhouse flock indicated their carnivorous specialization [60]. Enriched lactic acid bacteria in landfill flock enhance carbohydrate metabolism, which may also help the marabou storks live in sanitary-severe conditions, probably owing to lactic acid bacteria’s antagonism to prevent pathogen invasion.

#### 4.1.5. Reptiles

In natural habitats, reptilian diets primarily comprise arthropod prey [61]. The elevated gut microbiome diversity observed in urban reptiles, e.g., eastern water dragons (*I. lesueurii*), likely stems from dietary diversification [21,61]. Specifically, exotic plants in urban areas provide novel plant-based resources, potentially enhancing microbial diversity [95]. Parallel patterns occur in other taxa [61,96,97,98]. Nevertheless, urban reptiles do not universally exhibit higher diversity than their natural counterparts, as outcomes depend on species-specific ecological niches and diet. *Bradypodion setaroi* and *B. thamnobates* show minimal microbiome divergence between natural and urban habitats due to negligible vegetation differences [10].

This homogenization is pronounced in dietary opportunists. *Podarcis siculus* consumes fruits, lizards, carrion, and anthropogenic waste (e.g., cheese, pasta) [61]; similarly, *I. lesueurii* shifts from native prey (insects, vegetation, and small reptiles) to exotic plants and human-provided food in city parks [21]. Supporting evidence comes from elevated nitrogen isotope (measured by δ^15^N) levels, indicating higher animal protein intake from anthropogenic sources.

#### 4.1.6. Summary

Urbanization drives significant diet changes, which significantly reshapes the GM of both humans and wildlife. Urban wildlife exhibit compositional homogenization of GM across species due to reliance on anthropogenic food subsidies, as demonstrated in sparrows [29,71,78], ibises [74], and coyotes [35,99], mirroring patterns observed in human populations. This convergence is further evidenced by shared amplicon sequence variants (ASVs) observed among urban anoles, coyotes, sparrows, and humans. Specifically, the nutritional transitions restructure gut microbiomes through three primary pathways: (1) Diversity divergence: Dietary generalists (e.g., tree sparrows) show elevated α-diversity via macronutrient variability [59,100,101], while specialists (e.g., house sparrows) experience diversity loss under processed diets [1,29,102,103]. (2) Taxonomic restructuring: Increased abundance of Proteobacteria (e.g., urban sparrows) and bile-tolerant taxa (*Bacteroides* spp., *Alistipes* spp.) is associated with high-fat/sugar diets [1,83,104], contrasting with *Firmicutes* dominance in rural conspecifics metabolizing complex carbohydrates [59,105,106]. Gut bacteria (e.g., *Listeria* spp., *Enterococcus* spp., *Brucella* spp., *Alistipes* spp.) enzymatically transform primary bile acids via multistep modifications (deconjugation, 7α/β-dehydroxylation, oxidation/epimerization, reconjugation). Bile acids reciprocally regulate microbiota by promoting bile acid-metabolizing bacteria, suppressing bile-sensitive species, and activating receptors (nuclear farnesoid X receptor, G protein-coupled membrane receptor, vitamin D receptor, pregnane X receptor) [107]. (3) Functional adaptation: This transition from diverse, fiber-rich diets to uniform processed foods reduces microbial diversity and functional capacity [21,98,108,109], most prominently in urban generalist species exploiting anthropogenic food subsidies [17,35,110]. During fiber deficiency, mucin-degraders (*Akkermansia* spp.) expand [49,84], while butyrate-producers (*Erysipelotrichaceae* spp., *Lachnospiraceae* spp.) enhance energy harvest in urban birds and coyotes [111,112,113,114]. Collectively, diet is the dominant driver of gut microbial ecology in wildlife. Gut microbial reprogramming directly mirrors dietary upheaval. Rapid microbial structural shifts facilitate swift responses to urban diet [28,29,85], potentially aiding host acclimation to urban environments [29]. Critically, low-MAC diets drive irreversible microbial extinction in mammals, with effects compounded across generations [5,78].

### 4.2. Environmental Pollution

Urbanization, driven by industrialization, enhanced medical access, and elevated living standards, inevitably generates multifaceted environmental pollutants—including noise, airborne particulates, aquatic contaminants, antibiotic residues, anthelmintic or pesticides, and heavy metals. As host microbiota resembles local environmental microbiota [61], urbanization-altered environmental microbes could directly colonize animals. Subsequently, these collectively perturb GM composition and function in humans and wildlife [80]. Such pollutant-induced changes partly arise from adaptive responses to environmental stressors and are also considered as drivers of GM disorders, triggering pathophysiological cascades: compromised nutrient assimilation and metabolic efficiency [59]; impaired immune homeostasis with heightened susceptibility to pathogens (e.g., C. difficile colonization [115]); and systemic pathologies including metabolic disorders, neuroinflammatory conditions (e.g., Parkinson’s disease), and antimicrobial resistance-associated infections [116]. Ultimately, these alterations diminish cross-species host fitness while amplifying ecosystem-level health risks via One Health transmission cycles [80].

Urbanization intensifies noise pollution, altering GM and inducing animal sickness. For instance, shifts from *Proteobacteria* to *Actinobacteria* dominance occur, along with increased cecal *Roseburia* spp. abundance, potentially causing weight loss and glucose intolerance [117]. Hence, the noise-induced disturbance of the GM may mediate the progression from glucose imbalance to metabolic syndrome [117], while also disrupting the oxidant-inflamm-barrier function of the microbiota–gut–brain axis. Urban particulate matter (PM) induces physiological disorders linked to GM. PM exposure in mice elevates schizophrenia risk, increasing *Verrucomicrobia* abundance but reducing *Fibrobacteres* and *Deinococcus*-*Thermus*—taxa involved in choline metabolism and estrogen signaling. Fecal transplants from PM-exposed mice partially replicate schizophrenia-like behavior [118]. In humans, PM exposure and noise pollution correlate with impaired fasting glucose, type 2 diabetes risk, and reduced microbial diversity [119].

Synergistic effects of urbanization-derived pollutants—airborne particulates (PM_2.5_/PM_10_), pesticides (e.g., nitenpyram, permethrin), heavy metals (e.g., arsenic), and residual chemicals—drive gut dysbiosis through interconnected mechanisms: (1) Microbial diversity suppression: PM_2.5_ exposure significantly lowers α-diversity and *Firmicutes* and *Verrucomicrobia* abundance, as demonstrated in a large Chinese cohort (*n* = 6627) [120]. Urban chemicals, such as cleaning products and detergents, further diminish environmental microbiome diversity, restricting the colonization of beneficial gut taxa such as *Prevotella* spp. [121]. (2) Metabolic disruption: PM_2.5_ exposure increases *Coriobacteriaceae* spp. abundance, elevating phenylacetylglutamine levels, a biomarker linked to type 2 diabetes mellitus (T2DM) [122], and simultaneously reducing butyrate-producing bacteria, causing a 70% decrease in cecal butyrate and impairing pancreatic β-cell function of glucose regulation [123]. Pesticide exposure, particularly nitenpyram, disrupts branched-chain amino acid and tricarboxylic acid cycles by sharply increasing *Akkermansia* abundance (by 200%) and reducing lactobacilli populations [80]. (3) Pathobiont proliferation: Permethrin exposure reduces *Bacteroides* spp. abundance 3.8-fold while boosting *Enterobacteriaceae* spp. populations [80]; arsenic exposure shifts gut communities by increasing *Enterobacter* spp. and depleting commensal Clostridia, leading to gut barrier dysfunction [80]. Collectively, such gut microbiome dysbiotic shifts elevate fasting glucose impairment (PM_2.5_-associated OR = 1.24), T2DM risk, neuroinflammation through short-chain fatty acid (SCFA) imbalance, and NCD susceptibility (by 37% due to reduced microbial cross-colonization) [121].

## 5. Consequences of Urbanization-Induced Gut Microbiota Alterations

Gut microbial dysbiosis triggered by urbanization impacts host physiology in multifaceted ways, including impaired immunity, behavioral alterations, and heightened susceptibility to infections and NCDs. Specifically, dysbiosis impairs metabolic efficiency by disrupting vitamin biosynthesis, nutrient assimilation, and dietary energy harvest [111]. Reduced microbial diversity further undermines immune homeostasis through compromised immune maturation and diminished antimicrobial metabolite production, increasing vulnerability to pathogens (e.g., *Clostridium difficile*) [115]. This dysregulation propagates systemic pathologies via pathways such as the gut-brain axis and the gut–liver axis, manifesting as metabolic disorders (e.g., obesity), inflammatory conditions (e.g., asthma), type 2 diabetes, cardiovascular disease, and neurological impairments (e.g., Parkinson’s disease) [116]. Critically, these alterations cascade into diminished host resilience with ecosystem-wide consequences [61].

### 5.1. Physiological Alterations Drive Immunological Dysfunction and NCDs

Urbanization profoundly reshapes GM composition and functionality, initiating cascading physiological and metabolic consequences. Regarding body size, a significant positive correlation exists between lizard body mass and gut microbial α-diversity in *Podarcis siculus*, attributable to expanded intestinal volume and prolonged digestive transit time facilitating microbial colonization [61]. In reproduction, elevated *Corynebacterium* abundance in female *Podarcis bocagei* lizards correlates with diminished reproductive fitness, aligning with avian studies demonstrating that this genus impairs female reproductive performance [61,124]. For thermal adaptation, the gut microbiome directly modulates host thermotolerance [125]. Metabolically, urban gut microbiomes exhibit compromised proteolytic capacity—particularly in nucleotide/amino acid catabolism—restricting nutrient assimilation in wildlife [59,126,127,128], whereas lipid metabolism pathways are paradoxically upregulated via PPAR signaling, potentially as an adaptive response to urban dietary lipids [56,129,130,131]. This is evidenced by enrichment of *Lachnospiraceae* and *Lactobacillaceae* (taxa implicated in lipid digestion) in urban sparrows [14,59,126,127,128]. Collectively, these microbiota-mediated alterations disrupt host energy homeostasis, stress adaptation, and physiological robustness during urbanization, reflecting microbial regulation of physiological adaptations to urban environments.

Furthermore, urbanization-induced GM alterations modulate metabolite production, exerting profound immunomodulatory effects. In humans, rural-to-urban transition reduces microbial diversity and perturbs immunometabolic pathways; notably, *Bifidobacterium longum* and *Akkermansia muciniphila* mediated histidine/arginine metabolism. This dysregulates TNF-α/IFN-γ production, elevating susceptibility to inflammatory disorders [132]. Microbial metabolites, such as short-chain fatty acids and secondary bile acids, produced by intestinal symbiotic bacteria, critically regulate the gut-immune axis. Short-chain fatty acids produced by intestinal symbionts modulate innate immunity (e.g., NLRP3 inflammasome activation, TLR signaling, neutrophil/macrophage function, NK cell activity), and adaptive immunity (T/B cell differentiation), thereby influencing pathologies including allergic airway inflammation, colitis, and osteoporosis [133]. Secondary bile acids produced by GM regulate RORγ^+^ regulatory T-cell homeostasis [134]. Consequently, depletion of these metabolites exacerbates gastrointestinal infections, inflammatory bowel diseases, cardiometabolic disorders, autoimmunity, and oncogenesis [135]. In wildlife, anthropogenic diets enrich pro-inflammatory taxa (*Erysipelotrichiaceae*, *Coriobacteriaceae*, *Lachnospiraceae*) in urban coyotes, correlating with splenomegaly—a biomarker of chronic immune activation [35,136,137,138,139,140]. Notably, this microbiota-driven inflammation mirrors human pathologies, as shown by post-splenectomy *Lachnospiraceae* depletion [140], highlighting bidirectional organ–microbiome crosstalk.

Urbanization profoundly reshapes the gut microbiome, characterized by depleted α-diversity, loss of beneficial taxa (*Prevotella*, *Faecalibacterium*), and enrichment of pro-inflammatory pathobionts (*Bilophila wadsworthia*, *Enterobacteriaceae*) [115,121]. This dysbiosis impairs metabolite production and mucin barrier maintenance [115,141,142], thereby inducing intestinal inflammation through (i) diminished SCFA-mediated anti-inflammatory signaling; (ii) increased gut permeability enabling microbial translocation; (iii) elevated pro-inflammatory metabolites; and (iv) LPS-activated TLR/NF-κB pathways [115]. These cascades exacerbate insulin resistance and oxidative stress, elevating risks for obesity, T2DM, cardiovascular disease, and colorectal cancer [121]. Urban diets further amplify risks by enriching CRC-linked *Fusobacterium nucleatum* and impairing glucose homeostasis [121]. Concurrently, urbanization reduces microbial diversity and alters key taxa: African populations show declining *Prevotella copri*, which is associated with reduced cardiometabolic risk [143] and increasing *Akkermansia muciniphila*, linked to pro-inflammatory cytokine elevation [132,144]. Collectively, these alterations underscore urbanization as a major driver of global NCD burden [121,145].

### 5.2. Catalyzing the Spread of Zoonotic Pathogens

Urbanization and globalization profoundly reshape the global epidemiology of infectious diseases. Expanding urban–wildland interfaces and global mobility accelerate cross-species spillover risks by enhancing pathogen transmission between wildlife reservoirs and human populations [146,147,148,149]. According to WHO estimates, 61% of all human diseases have zoonotic origins, and 75% of newly identified pathogens (including SARS-CoV-2) are zoonotic in nature. This risk amplification stems from urban-driven mechanisms that disrupt wildlife gut microbiomes and intensify transmission pathways.

First, urban-induced gut microbiome dysbiosis, characterized by reduced microbial diversity, promotes pathogen colonization and increases wildlife susceptibility to opportunistic infections [80,150,151]. For example, urban diet shifts (high-sugar anthropogenic foods) increase pathogenic loads of *Yersinia* (yersiniosis), *Campylobacter* (campylobacteriosis), and multidrug-resistant *E. coli* (enterohemorrhagic colitis) in birds [78,80,152,153]. Urban coyotes consuming anthropogenic food exhibit gut microbiome shifts linked to reduced kidney fat assimilation, immune stress, and elevated risk of *Echinococcus multilocularis* infection (causing alveolar echinococcosis) [35]. Model analyses demonstrate that urbanization disrupts co-evolved host–microbe immune barriers, thereby facilitating parasite invasion [35,154]. Reptiles under urban warming and altered food resources show pathogenic microbial shifts, as observed in crocodile lizards and common lizards [61,155,156]. Pathogenic *Escherichia* spp. and *Shigella* spp. (bacillary dysentery) colonization in urban residents may amplify antibiotic resistance [55]. Moreover, urban sparrows in Arizona show increased *Pseudomonas* spp. (e.g., disease-causing *P. aeruginosa*) abundance [73]. *Campylobacteraceae* spp. surges in San Francisco’s urban white-crowned sparrows, potentially inducing human gastroenteritis [91].

Second, synanthropic wildlife (e.g., urban pigeons, sparrows) thrive and proliferate in densely populated cities, acting as reservoirs for zoonotic pathogens, including *Cryptococcus neoformans* (cryptococcosis), *Leptospira* spp. (leptospirosis), *Salmonella* (salmonellosis), and antifungal-resistant *Candida* spp. (candidiasis) [14,129,130,146,157]; these zoonotic pathogens pose substantial threats to human health, causing significant morbidity and mortality globally [131,158]. Critically, urban proximity amplifies spillover risks through direct contact (e.g., handling birds), inhalation of contaminated aerosols (e.g., pigeon droppings), or exposure to pathogen-laden environments [159]. Notably, urban pigeons in Porto Alegre show 100% carriage of *C. neoformans*, a key agent of fatal meningitis in immunocompromised humans [157,160]. Pathogenic bacteria like *Vibrio* spp. (cholera agent) are enriched in swans overwintering at Poyang Lake [161]. Furthermore, anthropogenic factors (e.g., inadequate waste management, antibiotic pollution) facilitate vector proliferation, promoting the spread of zoonotic pathogens. Antibiotic pollution in Indian rivers reduces waterfowl microbiome diversity, heightening H5N1 susceptibility [74,162]. Poor sanitation in Rio de Janeiro promotes *Aedes aegypti* expansion, exacerbating Zika/dengue outbreaks [129,157]. These changes drive contrasting urban–rural metabolomic and immune profiles, marked by key cytokines (TNF-α, IFN-γ, IL-1β), showing the strongest urban–rural divergence [132].

Urbanization also drives the proliferation of antimicrobial-resistant bacteria through profound gut microbiome alterations across species. Frequent antibiotic use in urban settings elevates antibiotic resistance gene (ARG) abundance in human gut microbiomes [57], while synanthropic wildlife (e.g., landfill-foraging gulls [*Larus* spp.] and crows [*Corvus* spp.]) develop ARG-enriched microbiomes that disseminate multidrug-resistant pathogens carrying *mdtG*, *emrD*, *mepA*, and *vanA* determinants [14,130,146,157]. Migratory species like white storks (*Ciconia ciconia*) further amplify ARG transmission (*sul1*, *blaTEM*, *tetW*, *blaKPC*, *mcr-1*) across continents [163,164]. Concurrently, wastewater contamination concentrates clinical ARGs (*sul1*, *blaTEM*, *qnrS*) in aquatic systems, which enter food chains via waterfowl ingestion [129,157]. Even soil ecosystems reflect this trend: enrofloxacin exposure reduces gut microbial diversity in earthworms while increasing pathogenic *Bacillus/Acinetobacter* and antibiotic-producing *Streptomyces*/*Nocardioides*—indicating microbiome-mediated self-protection responses that inadvertently select for resistance [165,166]. These interconnected alterations underscore the imperative for One Health surveillance to integrate gut resistome tracking and microbiome-resilient urban planning [14].

Together, these processes underscore how urbanization acts as a catalyst for zoonotic emergence, highlighting the need for integrated surveillance of urban microbiomes, wildlife health, and pathogen flow.

### 5.3. Microbiota-Mediated Behavioral Adaptations in Urban Wildlife

Accumulating evidence indicates that microbe–host interactions modulate not only health and disease states but also emotions, cognition, and behavioral patterns [37]. This relationship is particularly pronounced in urban wildlife, where anthropogenic stressors drive behavioral adaptations. Dysbiosis represents a key causal pathway through which urbanization shapes wildlife behavior. Innovations in foraging strategies (e.g., songbirds opening milk bottles [167]) exemplify cognitive adaptations to novel urban resources [168]. However, concomitant alterations in the microbiome may promote maladaptive behaviors, posing conservation challenges. For instance, a direct correlation exists between altered GM composition and aggression in canids: an urban coyote (*Canis latrans*) exhibiting fatal aggression lacked detectable *Fusobacteria* in its microbiome—a deviation from healthy conspecifics with robust prey consumption [35]. This finding aligns with associations between reduced *Fusobacteria* abundance and heightened aggression observed in domestic dogs. This linkage arises from diet-mediated feedback loops: protein-rich diets increase predominantly rural-associated taxa such as *Fusobacteria* spp., *Anaerobiospirillum* spp., and *Sutterella* spp., enhancing predatory efficiency; conversely, anthropogenic foods enrich lactic acid bacteria (*Streptococcus*, *Enterococcus*) while compromising body condition and foraging competence, thereby reinforcing dependence on human-derived resources. Such dysbiosis-driven cycles exacerbate human–animal conflicts by diminishing natural foraging behavior and escalating aggression during prolonged interactions [169]. Elucidating these mechanisms and identifying specific solutions are critical for designing cities that mitigate human–wildlife conflict while promoting biodiversity.

## 6. Suggestions for Improvement

The gut microbiome constitutes a fundamental pillar of host physiology, regulating intestinal barrier integrity, immune homeostasis, and environmental adaptation [59]. Urbanization disrupts evolutionarily conserved host–microbe symbiosis through dietary homogenization, antibiotic overexposure, and landscape fragmentation [59], inducing dysbiosis characterized by diminished microbial diversity and metabolic dysfunction. These perturbations compromise critical ecosystem services delivered by commensal microbiota, such as nutrient cycling and colonization resistance, while simultaneously increasing disease susceptibility. Industrialized populations exemplify this through syndemic NCDs in industrialized populations [59]. With two-thirds of the global population projected to inhabit urban areas by 2050 [69], urbanization’s detrimental effects on gut microbiomes will intensify across species. This trajectory underscores the urgency of implementing preemptive interventions during early urbanization phases [10]. Consequently, microbiome-targeted restoration strategies present an evidence-based approach to alleviate escalating biomedical and ecological burdens, thereby bridging conservation physiology and public health priorities [10].

For humans, dietary interventions represent a critical strategy to counteract urbanization-induced gut dysbiosis. Clinical evidence demonstrates that targeted diets—such as the Mediterranean diet (high in complex carbohydrates/polyphenols)—restore beneficial taxa (*Prevotella* spp., *Faecalibacterium prausnitzii*, *Bifidobacterium* spp.) and elevate SCFA concentrations, reinforcing gut barrier integrity and attenuating inflammation [121]. Similarly, a microbiome-restorative diet mimicking non-industrialized patterns in Canadian adults reversed industrialization-altered features (e.g., increased fermentation capacity, reduced mucus-degrading potential) and reduced pro-carcinogenic metabolites (deoxycholic acid, 8-hydroxyguanine), concurrently lowering low-density lipoprotein (17%), fasting glucose (6%), and C-reactive protein (14%) [115]. Conversely, ketogenic diets suppress obesity-associated microbes (*Lachnospiraceae* spp., *Blautia* spp.) and ameliorate colitis via microbiota-dependent immunomodulation (e.g., reduced RORγt+ ILC3, increased *Akkermansia*) [121]. Nevertheless, context-dependent microbial activities (e.g., mucin degradation by *A. muciniphila* under fiber depletion [121]) and the irreversibility of fiber deficiency-driven dysbiosis [121] underscore the need for preserving dietary macronutrient complexity. Fermented foods and ethnic-specific diets (e.g., Zang butter tea) further diversify microbiomes and reduce disease risks, though their mechanisms warrant deeper interrogation [121]. Additionally, probiotic supplementation represents a promising strategy to mitigate urbanization-driven gut dysbiosis by regulating the balance of GM, mucin synthesis, and tight junction proteins, competitively excluding pathogens, enhancing colonization resistance, and restoring microbial community stability [80].

However, for wild animals inhabiting cities, direct dietary and probiotic interventions are unfeasible. As urban expansion intensifies human–wildlife contact interfaces, zoonotic epidemic risks escalate. To mitigate urbanization-induced gut dysbiosis in wildlife and reduce zoonotic spillover, landscape-scale interventions restoring microbial resource availability are essential. We therefore propose the following evidence-based strategies: (1) Establish protected wildlife corridors to confine urban wildlife mobility and minimize human contact. These multi-taxa corridor networks should connect fragmented green spaces, facilitating microbial dispersal and genetic exchange among populations to enhance adaptive resilience [21]. (2) Enhance habitat heterogeneity by increasing vegetation cover and spatial configuration complexity. This diversifies foraging resources, promoting intake of microbiome-stabilizing microbiota-accessible carbohydrates (MACs) [14]. (3) Restrict wildlife access to anthropogenic food sources and environmental pollutants to maintain GM homeostasis and prevent pathogen colonization. Anthropogenic food buffer zones with native flora need to be created to reduce reliance on pro-inflammatory human-sourced foods (e.g., processed items), thereby counteracting Bilophila enrichment [170]. Concurrently, urban planning must prevent human-polluted water systems and waste facilities from encroaching on wildlife ecological corridors. (4) Implement systematic disease surveillance programs for urban wildlife populations. Critically, all interventions must balance taxon-specific needs (e.g., canopy cover for avian microbiomes versus open-ground patches for pollinators) to prevent ecological trade-offs [170]. Integrating these measures sustains host–microbe coadaptation, which is vital for wildlife phenomic plasticity under anthropogenic stress [21].

## 7. Conclusions

Urbanization is reconfiguring global biodiversity and infectious disease dynamics by altering human–animal–environment interfaces through habitat loss, fragmentation, and intensified human–wildlife overlap. This review documents urbanization-driven gut microbiome alterations across five amniote lineages (reptiles, birds, carnivores, non-human primates, humans) (Table 1 and Table 2), synthesizes evidence on urban dietary-pollutant drivers of wildlife microbiota reconfiguration, evaluates consequent host physiology–behavior–immunity remodeling, and analyzes microbiota-mediated zoonotic–antimicrobial-resistance–NCD impacts. We demonstrate that urbanization restructures gut microbiomes via taxon-specific mechanisms, with notable microbiome convergence between urban vertebrates (e.g., crested anoles, white-crowned sparrows, coyotes) and humans through anthropogenic food consumption and microbial exchange. While such plasticity may enhance host adaptability, the loss of conserved microbial taxa elevates risks of disease susceptibility, mortality, and reproductive impairment. Critically, the impacts of non-dietary urbanization factors (such as climate change and drought) on GM remain uncharacterized. The specific mechanisms underlying these microbial community modulations and their health consequences (particularly for wildlife) are poorly resolved, necessitating prioritized future research. Furthermore, standardized frameworks for assessing urbanization levels and quantifying specific urban factors remain lacking. For instance, while isotope labeling serves as a robust tool for dietary analysis, its practical application in urbanization research is limited. Strikingly, urban wildlife shares significantly more microbiota with humans than rural counterparts [24], indicating urban cross-species transmission pathways and convergent evolution under parallel selective pressures. We thus propose targeted interventions for urban residents and wildlife to mitigate microbiota-mediated harms. Integrating these advances within One Health frameworks will inform strategies to bolster wildlife resilience, reduce zoonotic risks, and safeguard biodiversity.

## Figures and Tables

**Figure 1 microorganisms-13-01736-f001:**
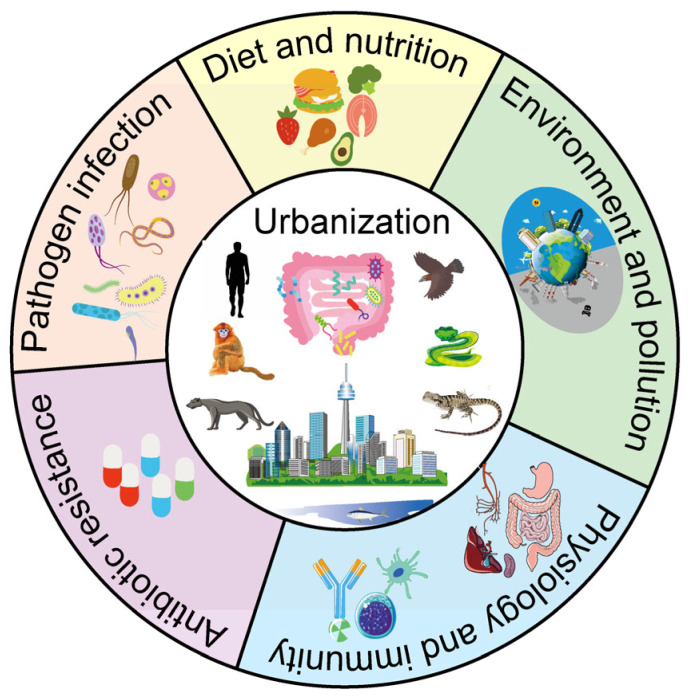
Exploring vertebrate gut microbial diversity in the urbanization era. Comprehensive synthesis across five key dimensions: (1) diet and nutrition, (2) pathogen exposure, (3) antibiotic resistance, (4) host physiology and immunity, and (5) environmental pollution.

**Figure 2 microorganisms-13-01736-f002:**
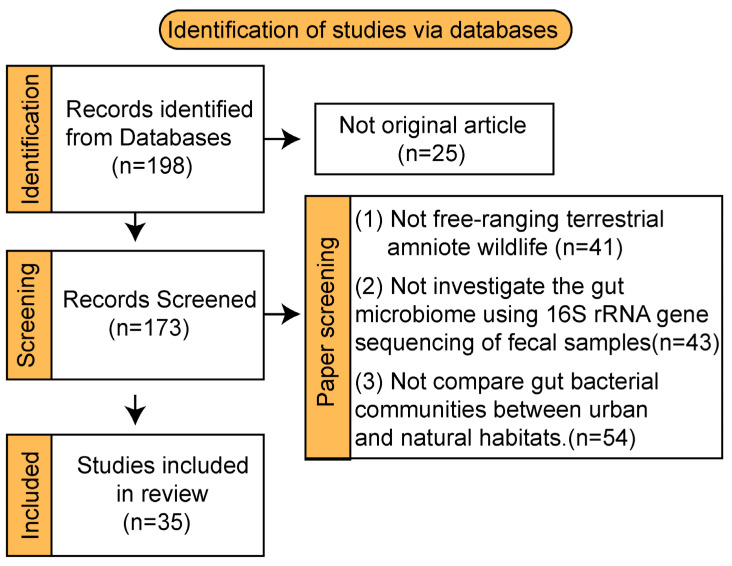
PRISMA flow diagram of study selection.

**Table 1 microorganisms-13-01736-t001:** Divergent human diet and gut microbiota across urban–rural regions.

Species	Time	Area	Country	Region	Food	Sample No.	Character of Gut Microbiome		Reason	Reference
							Bacteria	Diversity		
Kazakh Population	-	Urban	Kazakhstan	Astana, Karagandy, Kostanay, Ridder	Salt ↑, cholesterol ↑, protein ↑	502	*Firmicutes/Bacteroidetes* Ratios ↑, *Coprococcus* spp. ↑, *Parasutterella* spp. ↑	α, β ↓	1. Diet 2. Environment 3. Antibiotics 4. Lifestyle	[49]
		Rural		Torgay, Akzhar, Zhansary, Novodolinka	Carbohydrates, fiber	149	*Ligilactobacillus* spp., *Sutterella* spp., *Paraprevotella* spp.			
Yanomami	-	Urban	Brazil	Manaus	Industrialized foods	12	*Bacteroides* spp. ↑	ND	1. Antibiotics 2. Diets. 3. Lifestyle.	[43]
		Rural		Roraima, Amazonas	Seeds, roots, fruits, fish	18	*Prevotella* spp., *Lactobacillus* spp., *Treponema* spp. *, *Succinivibrio* spp. *			
Amaxhosa	November 2019–February 2020	Urban	South Africa	Cape Town	Energy, fat, animal protein diet	20	Bacteroidota ↑, Proteobacteria ↑, *Prevotella* spp. ↓ *Faecalibacterium* spp. ↓	α ↓	1. Diet 2. Lifestyle	[171]
		Rural		Zithulele	Fiber, plant protein, polyphenols	24	*Prevotella* spp., *Faecalibacterium* spp., *Dialister* spp., *Treponema* spp.			
Vietnamese	July 2013–2023	Urban	Vietnam	Hanoi	Westernized diet	40	Pathogenic, Bacteria (Enterobacteriaceae) ↑, *Bifidobacterium* spp. ↓.	α ↓	1. Diet 2. Breast milk composition 3. Environment 4. Lifestyle	[172]
		Rural		Tien Giang, Phu Tho, Ha Long Bay.	Carbohydrates, plant-based fat, animal protein	60	*Bifidobacterium* spp., Pathogenic Bacteria (Enterobacteriaceae)			
Malay	January 2019–October 2019	Urban	Philippines	Manila	Pasta, pizza, French fries, processed meat, mayonnaise, butter.	25	Clostridiales ↑	-	1. Diet	[173]
		Rural		Albay	Rice, starchy roots, green leafy vegetables, smoked fish, coconut milk	67	Prevotella-Driven Microbiome, Bacteroidetes, Proteobacteria			
Mongoloid	2018–2019	Urban	China	Ningxia Hui Autonomous	Meat ↑, rice ↑, potatoes ↓.	1204	*Blautia* spp. ↑, *Klebsiella* spp. ↑	α ↓	1. Urbanization level 2. Diet	[22]
		Rural			Potatoes, whole grains	1303	*Faecalibacterium* spp., *Prevotella* spp., *Pseudobutyrivibrio* spp.			
Black Race	2016–2017	Urban	South Africa	Soweto	Westernized diet	51	*Bacteroides* spp. ↑	α ↓	1. Environment 2. Diet 3. Lifestyle 4. Epidemiological	[174]
		Rural		Bushbuckridge	Traditional plant-based diet	119	*Prevotella* spp., *Vampirovibrio* spp.			
Amerindians	-	Urban	Mexico.	México City	Animal protein ↑, fiber ↓	13	Saccharibacteria *	α ↓	1. Diet 2. Lifestyle 3. Antibiotics 4. Hygiene	[175]
		Rural		Me’Phaa	Agricultural crops	29	Deinococcus-Thermus *, Chloroflexi *, Verrucomicrobia *			
Nigeria	-	Urban	Nigeria	Jos City	High-fiber foods	22	Bacteroidetes ↑, Spirochaetes ↑, *Prevotellaceae* spp. ↑;	α ↑	1. Diet 2. Lifestyle 3. Pathogens.	[42]
		Rural		Jengre Town	Low-fiber processed foods	28	Firmicutes, *Ruminococcaceae* spp., *Lachnospiraceae* spp., *Christensenellaceae* spp., *Blautia* spp.			
Caucasian	March 2017–April 2017	Urban	Italy	-	Mpd: vegetables, fruit, nuts, seeds, eggs, fish, lean meat Md: Mediterranean diet	158	Bile-Tolerant (*Bacteroides* spp., *Collinsella* spp., *Dorea* spp.) ↑, Fat-Loving Microbes (*Bilophila* spp.) ↑, SCFA Producers (*Lachnospira* spp., *Coprococcus* spp.) ↑	-	1. Diet 2. Environment 3. Lifestyle	[176]
		Rural	Tanzania, Peru, Canada	-	Hadza/Matses: High-fiber plant foods Inuit: Animal fat/protein	73	*Prevotella* spp.			
Tanzania and Botswana: African Black Usa: White, African American	Tanzania: March Botswana: April (No year)	Urban	USA	-	Industrial diet, fiber ↓.	12	Bacteroidaceae ↑	α ↓	1. Diet 2. Environment 3. Genetic relatedness 4. Lifestyle	[177]
		Rural	Tanzania, Botswana	-	Fiber.	114	Prevotellaceae			
Amerindian	January 2015	Urban	Venezuela	Caracas	Traditional rural diet.	7	*Bacteroides* spp. ↑, *Blautia* spp. ↑, *Faecalibacterium* spp. ↑	α ↓	1. Diet 2. Lifestyle 3. Environment 4. Exposure 5. Age difference:	[178]
		Rural		Bolivar, Venezuela	Mainly cassava, fish, various fruits, meat	38	*Treponema* spp., *Succinivibrio* spp., *Ruminobacter* spp.			
Thai People	-	Urban	Thailand	Bangkok	High-fat modern diet	17	Bacteroidales ↑, Selenomonadales ↑, Clostridiales ↓	α ↓	1. Diet 2. Lifestyle	[179]
		Rural		Buriram	Traditional vegetable-based diet	28	Clostridiales			
Leh:Mons, Mongols, Dards Ballabhgarh: Aryan Descendants.	-	Urban	India	Ballabhgarh	Processed foods ↑	24	Firmicutes ↑, Proteobacteria ↑, Lactobacillus spp. ↑	α ↓	1. Diet 2. Geographical locations 3. Lifestyle	[53]
		Rural		Ballabhgarh, Leh	(Ballabhgarh): Vegetarian (Leh): Non-vegetarian, low dairy intake.	60	Bacteroidetes, *Parabacteroides* spp., *Blautia* spp., *Prevotella* spp.			
Nigerian Ethnic Groups	July 2015–September 2015	Urban	Nigeria	Ilorin, Abeokuta, Ado Ekiti, Ibadan, Nigeria.	Traditional Nigerian foods and Western diet	30 (12 Infants, 18 Adults)	Firmicutes ↑, Proteobacteria ↑, *Firmicutes\Bacteroidetes* ↑. *Bacteroides* spp. ↑, *Bifidobacterium* spp. ↑, *Oscillospira* spp. ↑	ND	1. Environmental 2. Microbial dispersal 3. Diet 4. Lifestyle and healthcare practices	[44]
		Rural		Bassa	Self-sufficient diet of tubers, grains, untreated river water.	18 (9 Infants, 9 Adults)	Bacteroidetes, Spirochaetes			
Han Chinese	-	Urban	China.	Hunan Province	Westernized diet	20	Archaea ↓, *Escherichia* spp. ↑, *Shigella* spp. ↑	α ↓	1. Westernized diet 2. Hygiene practices 3. Antibiotics 4. Lifestyle	[55]
		Rural			Fiber, processed foods	20	Beneficial Bacteria (*Ruminococcus* spp.)			
-	2013	Urban	Italy	Bologna	Westernized diet	12	*Bifidobacterium* spp. ↑, Bacteroide ↑	-	1. Diet 2. Lifestyle 3. Environment	[180]
		Rural	Tanzania	-	Primarily plants (tubers)	17	*Prevotella* spp., *Succinivibrio* spp., *Treponema* spp., *Bulleidia* spp., *Bifidobacterium* spp., *Bacteroides* spp., *Blautia* spp., *Dorea* spp.			
Mongoloid	-	Urban	India	Port Blair	Carbohydrates, proteins	12	*Bifidobacterium* spp. ↑	α ↓	1. Diet 2. Lifestyle	[181]
		Rural		Nancowry	Agriculture, forests	12	Bacteroidetes			
Matses, Tunapuco, Residents Of Norman, USA	-	Urban	Peru	Norman, Oklahoma, USA	Processed foods, bread, dairy products	56	*Bifidobacterium* spp. ↑, *Ruminococcus* spp. ↑, *Blautia* spp. ↑	α ↓	1. Diet 2. Lifestyle	[182]
		Rural		Matses, Tunapuco	Tubers, fish, game meat, rare dairy, processed foods	23	Spirochaetes, Proteobacteria			
Hadza Hunter-Gatherers	January 2013–April 2013	Urban	Italy	Bologna	Mediterranean diet	16	*Bifidobacterium* spp. ↑	α ↓	1. Diet 2. Lifestyle 3. Environment	[38]
		Rural	Tanzania	-	Wild foods	27	Firmicutes, Bacteroidetes			

(*): Specific bacteria; ND: no difference; ‘-’: no data; ‘↑’: indicate an increase; ‘↓’: indicate a reduction; ‘α’: alpha diversity; ‘β’: beta diversity.

**Table 2 microorganisms-13-01736-t002:** Diet–gut microbiota divergence in non-human amniotes across urban and natural habitats.

Species	Time	Area	Country	Region	Food	Sample No.	Character of Gut Microbiome		Reason	Reference
							Bacteria	Diversity		
Birds										
*Anser cygnoides*	Poyang Lake: January 2015	Urban	China	Poyang Lake	Stems of submerged macrophytes	20	Proteobacteria ↑, *Clostridium* spp. ↑, Lactobacilli ↑, Basidiomycota ↑	α ↓	1. Environment 2. Human activities	[161]
	Khukh Lake: August 2014	Natural	Mongolia	Khuvsgul Lake		20	*Turicibacter* spp., *Solibacillus* spp., Ascomycota			
*Zonotrichia leucophrys*	2021	Urban	USA	California	Diets similar to humans	87	Similar to humans (*Bacteroides* spp. ↑)	α ↓	1. Bacterial spillover 2. Diet convergence 3. Environmental pressures	[28]
		Natural			Natural foods		Differ significantly from humans			
*Geospiza fuliginosa*	February 2018–May 2019	Urban	Ecuador	Provincia De Galápagos	More diverse diet, processed foods	44	*Firmicutes* spp. ↑, *Arthromitus* spp. ↑	α ↓	1. Diet 2. Urban physiological stress 3. Vertical transmission	[150]
		Natural			Insects	14	*Peptostreptococcus* spp. ↑, *Klebsiella* spp., *Erysipelatoclostridium* spp. ↑.			
*Leptoptilos crumeniferus*	September 2019	Urban	Uganda	Kampala	City garbage	80	Lactobacilli ↑	α ↑	1. Diet 2. Environmental adaptation	[60]
		Natural			Pig waste	20	*Peptostreptococcus* spp.			
*Parus major*	May 2018–July 2018	Urban	Poland.	Warsaw	Anthropogenic food	76	Enterobacteriaceae ↑	α ↓	1. Reduced tree cover density 2. Sound pollution 3. Distance to city center	[183]
		Natural			Natural foods	-	*Catellicoccus* spp., Microbacteriaceae, Pseudonocardiaceae, Carnobacteriaceae, Sphingomonadaceae			
*Eudocimus albus*	October 2015–March 2017	Urban	USA	Florida	Human-provided food	82	Proteobacteria ↑, Bacteroidetes ↑.	α ↓	1. Habitat 2. Diet 3. Environment	[74]
		Natural			Natural foods		Firmicutes, Cyanobacteria.			
*Passer domesticus*	October 2016–December 2016	Urban	USA	Arizona	Human-derived food waste	7	Proteobacteria ↑, Pseudomonadales ↑, Pseudomonas ↑	α ↓	1. Diet 2. Physiological adjustment 3. Microbiota shift 4. Environment	[73]
		Natural			Human-produced grains	13	Proteobacteria, Pseudomonadales, Pseudomonas			
*Passer montanus*	November 2021–January 2022	Urban	China.	Hubei	Human food residues	10	Proteobacteria ↑	α ↑	1. Diet variations 2. Environmental exposures 3. Microbiota adaptation	[59]
		Natural			Natural foods	10	Firmicutes			
Non-human primates										
*Microcebus griseorufus*	2013–2015	Urban	Madagascar	Miarintsoa	-	47	Veillonellaceae ↑	α ↓	1. Diet shifts 2. Habitat 3. Fragmentation, human–livestock contact	[36]
		Natural		Andranovao	Gum, fruits, insects.	113	Bacteroidaceae, Verrucomicrobia			
Carnivore										
*Canis lupus*	February 2023–November 2024	Urban	USA	Tennessee	-	211	Fusobacteria ↑	α ↓	1. Management 2. Environment 3. Hygiene	[184]
		Natural				135	*Ancylostoma* spp.			
*Canis latrans*	2017–2018 2018–2019 winter	Urban	Canada	Alberta	Anthropogenic food	30	*Streptococcus* spp. ↑, *Enterococcus* spp. ↑	α ↓	1. Diet 2. Immune system stress 3. Activity range	[35]
		Natural			Natural prey, fruits	65	Fusobacteria, *Sutterella* spp., *Anaerobiospirillum* spp.			
Reptile										
*Bradypodion melanocephalum* *Bradypodion thamnobates* *Bradypodion setaroi*	-	Urban	South Africa	Ethekwini, Stlucia, Howick	Insectivorous animal	10	Firmicutes ↑, Desulfovibrionaceae ↑, Ruminococcaceae, Christensenellaceae ↑	ND	1. Similarities between urban and natural habitat vegetation	[10]
		Natural				10	Proteobacteria, Bacteroidota			
Rural: *Podarcis bocagei*, *podarcis lusitanicus* Urban: *Podarcis siculus*, *podarcis virescens*	September 2020	Urban	Portugal	Lisbon	Hemiptera, coleoptera, Diptera, Hymenoptera, and Araneae	41	*Odoribacter* spp. ↑, *Corynebacterium* spp. ↑	α ↑	1. Species influenced gut bacterial community structure only in lizards from the urbanized environment	[61]
		Natural		Moledo		61	*Corynebacterium* spp.			
Urban: *Anolis cristatellus*, Rural: *Anolis cristatellus*	July 2019	Urban	USA, Canada	Edmonton, San Francisco, Mayagüez	Podarcis virescens: more versatile diet, fruits and nectar, class Arachnida, and orders Hymenoptera, Hemiptera, Coleoptera, and Diptera Teira dugesii: insects, small fruits	127	*Bacteroides* spp. ↑, *Firmicutes/Bacteroidetes* ratio ↑	α ↓	1. Acquire GM associated with urban humans 2. Convergence of urban GM	[28]
		Natural		Maricao, Quemado, Leduc			*Bacteroides* spp.			
*Intellagama lesueurii*	November 2014–April 2015	Urban	Australia	Queensland	Insects, native vegetation, and small reptiles	41	*Ruminococcus* spp. ↑, *Lactobacillus* spp. ↑	α ↑	1. Diverse diet 2. Higher in fat	[21]
		Natural				25	*Blautia* spp. ↑, *Citrobacter* spp. ↑			

GM: gut microbiome; ND: no difference; ‘-’: no data; ‘↑’: indicate an increase; ‘↓’: indicate a reduction; ‘α’: alpha diversity.

## Data Availability

No new data were created or analyzed in this study. Data sharing is not applicable to this article.

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
