# Peer review of "Double-Edged Sword: Urbanization and Response of Amniote Gut Microbiome in the Anthropocene"

_microorganisms, 2025, doi:10.3390/microorganisms13081736_

Round 1
Reviewer 1 Report
Comments and Suggestions for Authors
In the manuscript by Peng et al. the authors summarise the knowledge on gut microbiota relative to five aspects: diet and nutrition, pathogen infection, antibiotic resistance, physiology and immunity, and environment and pollution. The five categories are well chosen and interesting to analyze however, the way the analysis is done seems superficial. More information is needed especially in the first chapter related to the effect of nutrition on gut microbiota. Here the selection of categories seems random, sometimes it is about the type of diet, other times about the group of organisms. Also, although the subchapter focuses on a type of organism other that seem completely unrelated are also presented as for example in the reptile subchapter foxes are also presented, mice are presented in the subchapter about birds. This mixing of different types of organisms and types of diets makes the information confusing and the conclusions are not clear. Also, it is not clear why the specific organisms were selected as the most relevant for that category, why water dragons for reptiles or coyotes for carnivores?
Each idea presented, sometimes a whole subchpater, seems to be the conclusion of one single study. As this is a review more data should be presented, compared, and contrasted. There is only one instance where contradictory information is presented, and even this is only just mentioned without any explanation on which is more likely to be the true conclusion. It is surprinsing that not more contradictory information has been found, unless this has been ignored.
The article needs to be completely rewriten, the main chapters are fine, their overall idea great but the presentation needs to be worked on and amount of information is very sparce and needs to be upgraded.
Some minor comments:
Some titles have typing mistakes in the small and capital letters
Figure 1 is referenced in the text but is not present in the manuscript. Please provide the referenced figure.
Reviewer 2 Report
Comments and Suggestions for Authors
The paper examines how urbanization affects the diversity and composition of the gut microbiota in vertebrates, with implications for adaptation, health, physiology and antibiotic resistance. By analyzing diet, pathogens, antibiotic resistance, immunity and pollution, a complex and often species-specific impact is highlighted. The work is extensive and well documented, but it should be improved in clarity and synthesis. It is recommended to strengthen the concluding part with critical insights and greater integration between the cases analyzed.
- Lines 45, 49, 170, 208, 249, 271 show Fig.1, however, there is no figure in the text.
- I consider the introduction too short. I would advise the authors to expand it by perhaps describing the gut microbiota in general, its functions, composition etc., in order to make the manuscript broader and more interesting. I recommend seeing: DOI: 10.3390/nu17060948
- Line 79-83/108-113/118-122/139-142/186-191/210-214/215-219/238-240/250-263/270-276/302-307 lacks a bibliographical reference, I would recommend adding it.
- Paragraph 2.3 is very interesting, however it is somewhat approximate. I advise the authors to expand on the insights in the text. For example, ‘Megamonas and Oscillospira are found exclusively in the gut microbiota of urban populations, which could be aô€„´ributed to the gradual urbanisation of these places and the adoption of a Western lifestyle’, this part is interesting, but lacks an explanation. Why are these bacteria associated with a Western lifestyle?
- I would recommend putting a summary table after paragraph 2, the data being many and also complex, in order to make the manuscript clearer and more readable.
- Paragraph 3 describes how pathogens vary in the gut microbiota, however, it is not explained what diseases these bacteria are associated with, nor why they are pathogenic. I would advise the authors to expand this part a little.
- The link between microbial changes and zoonotic risk for humans is mentioned, but not developed in detail. I suggest to deepen with a paragraph dedicated to the health implications for humans and zoonotic spillover risks in order to improve the manuscript.
- Line 208-209 the abbreviation ‘antibiotic-resistant bacteria (ARB)’ is used twice
- There are capital letters in the title of paragraph 6
- The conclusions are too brief, lack a critical analysis of the data described, and I also recommend adding some molecular explanation of why there is variation in the gut microbiota under certain conditions.
- I would advise the authors to add a ‘methods’ section where they describe the criteria used to select the selected papers, perhaps using a PRISMA diagram.
- Some studies seem to be cited only descriptively, without critical analysis of methods or limitations. My suggestion is to add a more thorough discussion of methodological biases of the reported studies (e.g. observational vs experimental studies, geographical differences).
- No reference to limitations of current literature or methodological challenges, I would recommend adding them.
Reviewer 3 Report
Comments and Suggestions for Authors
Dear Authors, here are my comments:
Lns 45,49,170, 208, 249 Fig 1 is missing.
Lns 46,50,61, etsc... Please explain what is ''urban diet''?
Ln 53 Please explain what is δ15N? Also δ13C, δ13N (Ln 68) and low-MAC (Ln 127).
Ln 62, 105, 107,146, 201 etc... Please use italic for Latin genera
Ln 237, 268 No need for world written in all capital letters
Ln 340 The term ''correlation'' is unsuitable, because you didn't use any of statistical tools to prove if there is any. Please rephrase.
The reference style for the MDPI is wrong. No need for doi numbers.
This is a very interesting topic. However, although the concept of the work is clear and precisely defined, in my opinion, it has not been adequately and completely realized. There is a lack of a clearer structure, a clearer connection of sentences into a logical whole, regional geographical similarities between e.g. upper and lower hemispheres, in addition to air and water pollution, it is necessary to address the impact of climate change and drought. Furthermore, there is no mechanistic explanation of these phenomena, only studies in which the biodiversity of intestinal microbiota in humans, birds and mammals is reduced/increased are listed. The conclusion is clear, but this should have been made clearer in the entire text.
The conclusion is that the goal of this interesting paper could have been realized much more clearly, with a clear and precise text within otherwise not bad division of the subtitles. Several tables and/or graphs are also missing, so that we could follow the text of the paper much more easily. Finally, the topic of urbanization could be approached a little more critically.
Comments on the Quality of English LanguageThe English could be improved to more clearly express the research. Please go through the entire text.
Reviewer 4 Report
Comments and Suggestions for Authors
microorganisms-3700213-peer-review-v1
Paper is an interesting and touching a very important topic but prepared in quite a superficial manner without really going to the diversity of examples, deeper discussion and comparison of the results.
Paper will need a very extensive extension, where authors will need to provide more examples and compare the facts between different authors. Moreover, almost all sections needs an extensive extension and more facts. In current way, authors just point on some facts, limiting their arguments to one or two examples and jumping to the next topic. Maybe help from more experienced colleagues will be a good way to improve the quality of the collection of data, interpretation and presentation.
Citations needs to be according to instruction form Publisher and Journal, as [1] and not as exponential symbols.
Figure 1 is not provided.
Maybe more illustrative (not only one figure, that in fact was not included) material will need to be provided.
Section on Ln46-49 need to be extended. In current way is a just 2 sentence general statement.
Section 2.1. was stated as reptiles, however, after the foxes been included in the section. This two groups needs to be separated, and authors will need to provide more details regarding explored topics. In fact, just providing 6 lines about the Eastern water dragons (however, in other sources named Australian water dragon) and just 4 lines about foxes is a quite telegraphic. Moreover, information about red foxes do not have any reference. Authors will need to extend the information mentioned with some scientific facts regarding the topic.
Similar comments are for the following section, 2.2.
Authors need to pay attention regarding the use of the terminology regarding former genus Lactobacillus. Since 2020 the genus has been divided into 23 new genera. According to the paper published in 2020 (Zheng et al., 2020), when referring to the former genus Lactobacillus, it was recommended to use English term "lactobacilli", written without English and with not capital L. Please, consult the above-mentioned paper.
Ln83: any reference?
Ln106-113: any references?
The authors providing very brief information and some limited facts regarding changes in the microbiota of different animals and human groups, however, just limited to one or maximum 2 examples without really going deeper into the facts and discussion.
References needs to be formatted according to the instructions form Publisher and Journal.
Round 2
Reviewer 1 Report
Comments and Suggestions for Authors
The authors did a good job on rewriting and restructuring the manuscript. I find it is improved compared to the previous version. The chapters are better organized and follow a logical flow. Thank you for providing the figure and also including tables, they are useful. I deem the manuscript ready for publication, however I advise the authors to read it carefully as there are grammar mistakes and typos. Below are a few examples:
Line 155, should CM be GM?
Line 308 “with more pronounced shifts fed urban diets”, it seems something is missing from this sentence. Maybe it should read “with more pronounced shifts in rural birds fed urban diets”?
Sentence in lines 322-325 says a study compared the GM, but it does not say what it compared it to. The way the sentence is phrased makes it confusing.
Should stocks in line 327 be storks?
Is the word dietary used instead of diet? Dietary is an adjective and it should define a noun. In several instances the word dietary is used by itself leaving me to wonder what it describes. Some examples are line 303 and 336 where the sentences end in the word dietary, also line 346 “Urbanization drives significant dietary” – dietary what? The expression is unfinished.
Brackets open in line 341 are not closed.
Lines 350, 470, 555 there are two full stops at the end of the sentence, delete one
In some chapters/sentences references are superscript, while in others they are no longer superscript. Please make it uniform across the manuscript.
Line 370 “Collectively, diet as the dominant driver of gut microbial ecology in wildlife.” I believe as was meant to be is, otherwise the sentence has no verb.
Line 382 word arise is repeated twice, delete the first
Line 383 word and is repeated twice, delete the second
Line 400, who correlates?
Line 426 infectious (which is an adjective) should be replaced by infections (noun)
Line 440 word “digesta” is not a real word, please correct it.
Some words are written in capital letter, and they should not be; on example Secondary in line 459 (there are others)
In line 460 the sentence starts with small letter and it should be capital
Line 486 the word “underscore” is repeated, delete the second
Reviewer 2 Report
Comments and Suggestions for Authors
The authors adequately answered all questions.
Author Response
Comments: The authors adequately answered all questions.
Response: Thank you for your help for improving our manuscript.
Reviewer 3 Report
Comments and Suggestions for Authors
The Authors have significantly improved their manuscript.
The topic of this manuscript is extremely important from the ecological point of view, not only for researchers, but also for the general population. The manuscript is well written, with understandable sentences. The connection with the gut microbiome is especially interesting.
After careful reading of the new version, the Authors did splendid work to improve this manuscript. The subsections are adequate, well-structured and comprehensively written (not sure about whether the content should be removed, but I am okay if the Content stays after the abstract).
Conclusion is excellent. Please uniform the reference style (no need for superscript). Table 1 and Table 2 are excellent, very informative. Please check one more time for consistency in font and style formatting within the Tables.
Reviewer 4 Report
Comments and Suggestions for Authors
microorganisms-3700213-peer-review-v2
The paper was significantly improved. The original version was very basic and reported on limited information. However, authors have taking very seriously criticism from the original version and significantly improved the manuscript. In fact, the new version (Revision 1) can be considered with a completely new, improved version of the topic explored. In the current version authors have provided a complex version, improved in all senses and paper that can be suggested for publication. In my opinion authors have significantly improved the manuscript and in the current version is suitable for publication.
However, some points regarding formatting need to be taken into consideration by the authors.
Some examples:
Sections under 3.1-3.6 will be positive if they can be extended a bit more. Yes, some data was presented in Table 1, however, if some discussion on these data will be provided as text under these section, this will enrich the sections.
Please, be sure that “spp.”all around the text is not in italics
In some cases, bacterial names are without italics. Please, check the entire manuscript and correct it.
References need to be formatted according to the instructions from Publisher and Journal.
